# Secondary structure prediction for RNA sequences including N⁶-methyladenosine

Elzbieta Kierzek [1✉], Xiaoju Zhang[2], Richard M. Watson[2], Scott D. Kennedy [2], Marta Szabat[1], Ryszard Kierzek[1] & David H. Mathews [2✉]

There is increasing interest in the roles of covalently modified nucleotides in RNA. There has been, however, an inability to account for modifications in secondary structure prediction because of a lack of software and thermodynamic parameters. We report the solution for these issues for N⁶-methyladenosine (m⁶A), allowing secondary structure prediction for an alphabet of A, C, G, U, and m⁶A. The RNAstructure software now works with user-defined nucleotide alphabets of any size. We also report a set of nearest neighbor parameters for helices and loops containing m⁶A, using experiments. Interestingly, N⁶-methylation decreases folding stability for adenosines in the middle of a helix, has little effect on folding stability for adenosines at the ends of helices, and increases folding stability for unpaired adenosines stacked on a helix. We demonstrate predictions for an N⁶-methylation-activated protein recognition site from MALAT1 and human transcriptome-wide effects of N⁶-methylation on the probability of adenosine being buried in a helix.

[1] Institute of Bioorganic Chemistry Polish Academy of Sciences, Noskowskiego 12/14, 61-704 Poznan, Poland. [2] Department of Biochemistry and Biophysics and Center for RNA Biology, School of Medicine and Dentistry, University of Rochester, 601 Elmwood Avenue, Box 712, Rochester, NY 14642, USA. ✉email: Elzbieta.Kierzek@ibch.poznan.pl; David_Mathews@urmc.rochester.edu

It has long been appreciated that covalent modification of RNA is used by nature to expand the chemical repertoire of the four common nucleotides. tRNAs, in particular, are known to have prevalent modifications, and the roles of some of these have been elucidated[1]. For mRNAs and long non-coding RNAs (lncRNAs), it had been harder to identify sites of modification until recently when methods were developed using next generation sequencing technologies to identify modifications[2,3]. Modifications including deamination to inosine[4], pseudouridylation[5–7], 5-methylation of cytosine[8], and N[6]-methylation of adenosine[9–14] can now be localized transcriptome-wide.

N[6]-methyladenosine (m[6]A) is considered the most prevalent modification in mRNA, and m[6]A is also widespread in lncRNAs[15,16]. It is known to have writers that apply the modifications to specific positions (methyltransferases including METTL3 and METTL14), readers that identify sequences with N[6]-methylation (RNA-binding proteins including YTHDF2 and the YTH family), and erasers that can remove the modification (demethylases including FTO and ALKBH5), restoring the base to adenine[17,18]. Furthermore, there are hundreds of sites for which the m[6]A modification consensus site is conserved between the mouse and human transcriptomes[12]. The impacts of N[6]-methylation are being elucidated[19,20]. For example, N[6]-methylation is known to cause structural switches[21] and to expose protein binding sites that are otherwise not available for binding[22]. Additionally, m[6]A can regulate splicing[23].

RNA secondary structure prediction is in widespread use to help determine structure-function relationships[24,25], but has not been generally available for understanding the roles of covalent modifications[26]. For unmodified sequences, secondary structure prediction has been used to identify microRNA binding sites[27], design siRNAs[28,29], identify protein binding sites[30], and discover functional RNA structures[31,32]. These types of calculations have not been able to account for modifications without extensive user intervention because a set of nearest neighbor parameters are needed for estimating the folding stability of structures that include modifications[26,33]. A number of studies demonstrated an impact on folding stability by modifications[34–39], but no complete set of parameters have been available for RNA folding, as there are for RNA folding with the four prevalent bases[33]. At the same time, no software has been available for handling a larger alphabet of sequences containing modifications. This led to a chicken-and-egg problem; without software, there was no impetus to assemble parameters and without parameters there was no reason to write the software.

In this work, we developed a full set of nearest neighbor parameters for a folding alphabet of m[6]A, A, C, G, and U nucleotides. These parameters account for helix and loop formation, and they are based on optical melting experiments for 32 helices with m[6]A-U base pairs and 13 oligonucleotides with m[6]A in loop motifs. We also modified the RNAstructure software package to accept user-defined folding alphabets and to read and utilize thermodynamic parameters for these extended alphabets[40]. Together, these advances allow the prediction of RNA secondary structures for sequences with m[6]A. We demonstrate, for calculations with human mRNA sequences known to contain m[6]A, that N[6]-methylation alters the folding landscape so that m[6]A is less likely to be buried in a helix, i.e., stacked between two base pairs. We also provide a model for the change in protein binding affinity caused by N[6]-methylation of the lncRNA metastasis-associated lung adenocarcinoma transcript (MALAT1).

## Results

### Overview of methods

Secondary structure prediction for RNA sequences including m[6]A requires both a set of nearest neighbor folding parameters and software capable of using the set of

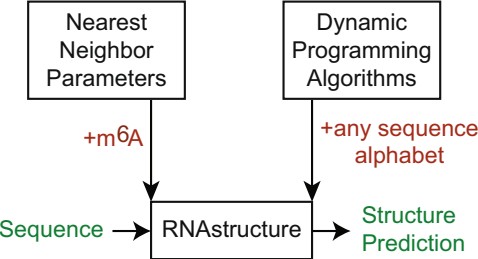

**Fig. 1 Overview.** In this study, we advanced the RNAstructure software package[40] (at center) to be capable of predicting secondary structures for sequences with the m[6]A nucleotide. RNA secondary structure prediction by RNAstructure relies on nearest neighbor parameters for estimating folding stability and dynamic programming algorithms for estimating structures and base pair probabilities. Here we fitted nearest neighbor parameters for m[6]A to optical melting data and revised software to be capable of considering any user-specified sequence alphabet.

parameters. An overview of the methods is illustrated in Fig. 1. RNA secondary structure prediction requires both parameters for evaluating folding stability and a search algorithm to identify the optimal structure given the parameters[24,25,41]. In our RNAstructure software, we use nearest neighbor parameters to estimate folding free energy change[33] and a set of dynamic programming algorithms that predict optimal structures[24,25].

We built a database of optical melting experiments of oligonucleotides including m[6]A and then used linear regression to fit nearest neighbor parameters. We also extended the functionality of RNAstructure[40] to recognize modified nucleotides in sequences and to use parameters for sequence alphabets beyond the four common nucleotides. The m[6]A modification parameters take advantage of this additional feature.

### Helix nearest neighbor parameters for m[6]A

The full set of Turner nearest neighbor rules for estimating RNA folding stability are based on optical melting experiments of 802 oligonucleotides and use 294 parameters[33,42]. We have shown, however, that the precision of a subset of parameters is more important than others for the precise prediction of secondary structure[43]. Following that work, we focused our experiments on estimating parameters for helices, dangling ends, and terminal mismatches.

Our first goal was to fit the 15 stacking nearest neighbor parameters for m[6]A-U pairs adjacent to Watson–Crick pairs, G-U pairs, or m[6]A-U pairs. For this study, 29 fully helical duplexes containing m[6]A-U pairs were synthesized and optically melted. This provides a total database of 32 fully helical duplexes with m[6]A-U base pairs. Supplementary Table S1 provides the duplexes and the stabilities determined by optical melting. These specific oligonucleotide sequences were chosen, in part, because analogous model RNA helices with A in the m[6]A position had been previously studied by optical melting (with the exception of GGUUAACC_2). This allows us to directly compare the folding stability with and without N[6]-methylation. We calculated the change in folding stability ($\Delta\Delta G°_{37}$) per methylation as compared to the unmethylated duplex. Supplementary Fig. S1 shows that the $\Delta\Delta G°_{37}$ is highly dependent on the adjacent sequence, ranging from +2.1 to −0.1 kcal/mol per methylation where positive free energies are destabilizing for methylation. Therefore, to estimate folding stabilities for duplexes with m[6]A-U pairs, a full nearest neighbor model is needed to account for the sequence dependence.

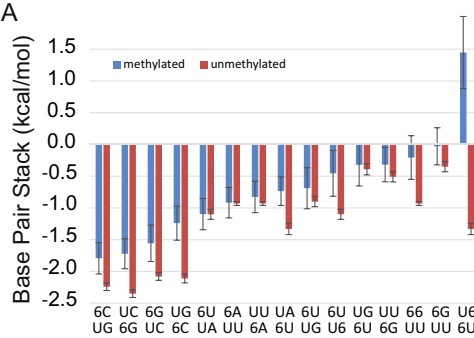

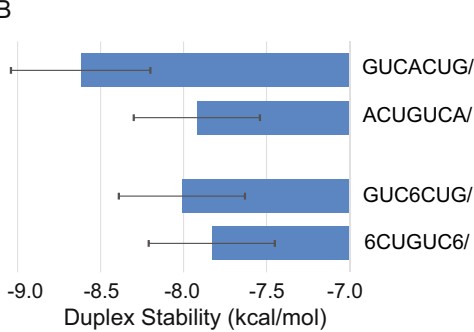

**Fig. 2 A comparison of base pairing stability for m⁶A to A. A** The nearest neighbor parameters for helix stacks. The position of the m⁶A is indicated by 6. The stacking parameters, determined by linear regression, are compared for methylated (blue; i.e., m⁶A-U base pairs) and unmethylated (red; i.e., A-U base pairs) sequences for analogous nearest neighbors. The unmethylated stacks (i.e., A-U base pairs) are those of Xia et al.[44] for adjacent Watson–Crick pairs and those of Chen et al.[79] for adjacent G-U pairs. Stacks with m⁶A-U pairs are generally less stabilizing than analogous stacks with A-U pairs. Uncertainty estimates are the standard errors of the regression. **B** Terminal m⁶A-U pairs are not destabilizing. Plotted are the duplex stabilities as folding free energy change from the linear fit to the $T_M^{-1}$ vs. $\ln(C_T/a)$ plots of the optical melting data. The top two sequences (Watson–Crick paired with a complementary strand) have the same nearest neighbor stacks, but the second helix has two terminal A-U pairs[44]. This costs 0.7 kcal/mol of stability. The bottom two sequences also have the same nearest neighbor stacks, but the second has two terminal m⁶A-U pairs. Here the stability cost is 0.18 kcal/mol and not outside of the uncertainty estimate, which is approximated as 4% of the total free energy change[44]. On average, terminal A-U pairs cost 0.45 kcal/mol of stability[44], but terminal m⁶A-U pairs are not destabilizing.

Linear regression was used to fit the nearest neighbor parameters for folding free energy change. Figure 2A shows the increments in comparison to the same stack with A-U pairs and Supplementary Table S2 provides the values. The free energy changes range from −1.79 ± 0.25 kcal/mol to +1.45 ± 0.57 kcal/mol. As expected based on prior optical melting experiments for duplexes with m⁶A-U pairs[34,36], nearest neighbor stacks for methylated A-U pairs are less stable than stacks for unmethylated A-U pairs. On average, the stacks with m⁶A-U pairs are 0.4 kcal/mol less stable per methylation. There are exceptions, however; an m⁶A-U pair followed by a U-A pair is as stable as an A-U pair followed by a U-A pair (−1.10 kcal/mol). The most unstable stack has two m⁶A-U pairs. Like A-U pairs, when the m⁶-U pair is adjacent a G-C it is more stable than when adjacent to A-U. Also like A-U pairs, m⁶A-U pairs adjacent to G-U are less stable than those adjacent to A-U pairs.

An unexpected feature of terminal m⁶A-U pairs is that they require no terminal penalty, although terminal A-U pairs receive a +0.45 ± 0.04 kcal/mol penalty per A-U pair at the end of a helix[44].

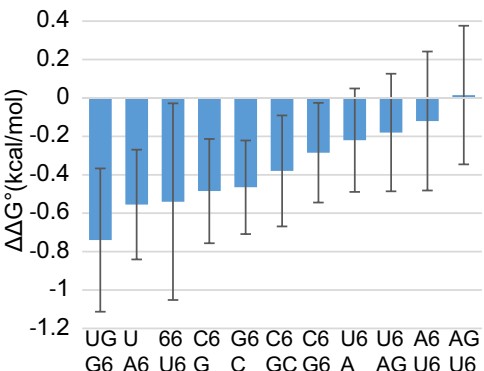

**Fig. 3 M⁶A stacking on a helix end stabilizes secondary structure as compared to A stacking.** The ΔΔG°₃₇ (kcal/mol) for dangling ends and terminal mismatches as a result of N⁶-methylation (Supplementary Table S4) is shown, where negative values mean greater folding stability for m⁶A than A. The motifs shown here have a terminal base pair (left side of motif), and either a dangling end or terminal mismatch right (right side of motif). On average, the methylated motifs are more stabilizing than the unmethylated motifs, although the extent of the stabilization is sequence dependent. Uncertainty estimates are propagated from the uncertainty from the individual optical melting experiments (see "Error propagation" in the "Methods").

Two findings support this. First, when a terminal parameter is included in the linear regression fit, the value is +0.13 ± 0.17 kcal/mol, which is not significantly different from 0 kcal/mol. Second, our dataset includes two helices with the same nearest neighbor stacks, but with different helix ends (Fig. 2B). Previously, it was noted that this pair of helices, when unmethylated, had markedly different stability (0.70 ± 0.47 kcal/mol), with the helix with A-U ends less stable[44]. For the methylated helices, the difference is small (0.18 ± 0.45 kcal/mol). This demonstrates that a terminal m⁶A-U base pair has overall similar stability to a terminal A-U base pair because a terminal A-U pair has a more favorable stack but requires the terminal A-U penalty.

**Loop nearest neighbor parameters for m⁶A.** For secondary structure prediction, parameters need to also be extrapolated for loop formation. The stability of a 3′ dangling m⁶A had been previously measured[34]. Additional optical melting experiments were performed in this work for two m⁶A 3′ dangling ends, an m⁶A 5′ dangling end, and seven terminal mismatches involving at least one m⁶A. One hairpin loop was measured with an m⁶A in the loop and not adjacent to the helix end. One 2 × 2 internal loop was measured with symmetric tandem G-m⁶A pairs. The loop sequences were chosen such that analogous sequences with A instead of m⁶A had been previously studied, so that the effect of methylation on stability can be quantified. Supplementary Table S3 provides the measured stabilities for these model structures and Supplementary Table S4 shows the stability of the loop motif in comparison to the motif with A.

As shown in Fig. 3, an m⁶A as a dangling end or as a component in a terminal mismatch stabilizes secondary structure formation to a greater extent than an analogous A. On average, the m⁶A dangling end is −0.43 ± 0.15 kcal/mol more stable than the analogous A dangling end for the 3′ and 5′ dangling ends studied here. Terminal mismatches for m⁶A-m⁶A, G-m⁶A, m⁶A-G, and m⁶A-C on Watson–Crick or G-U terminal pairs are on average −0.28 ± 0.26 kcal/mol more stabilizing than the analogous A-A, G-A, A-G, or A-C terminal mismatches. This stabilizing effect is sequence dependent; the ΔΔG°₃₇ ranges from −0.74 kcal/mol (G-m⁶A mismatch on a U-G pair) to +0.02 kcal/mol (G-m⁶A

mismatch on an A-U pair). An m⁶A-m⁶A mismatch on an m⁶A-U pair is more stable than the m⁶A-m⁶A mismatch on an A-U pair by $-0.42 \pm 0.40$ kcal/mol.

The hairpin loop structure with m⁶A is marginally less stable than the analogous hairpin loop with A ($\Delta\Delta G°_{37} = 0.23 \pm 0.24$ kcal/mol; Supplementary Table S4). The $2 \times 2$ internal loop with tandem G-m⁶A pairs is also marginally less stable than the analogous loop with tandem G-A pairs ($\Delta\Delta G°_{37} = 0.33 \pm 0.53$ kcal/mol; Supplementary Table S4). Both stability changes are within the uncertainty estimates, suggesting that they are not significant differences.

**Additional experiments to test the parameters**. To test our parameters, we performed additional melts of duplexes (Supplementary Table S5A). The first is a duplex with all base pairs, incorporating a consensus N⁶-methylation site, GGACU, where we determined the helix stability with and without methylation. The second is an additional 3′ dangling m⁶A to test our assumption that dangling m⁶A are stabilized by $-0.3$ kcal/mol compared to dangling A. Then we studied six duplexes containing bulge loops, with and without methylation, four of which are loops that were previously studied[45] and one of which models the m⁶A site in the hepatitis C viral genome[46]. In these bulge loops, m⁶A is the bulged nucleotide, in the base pair closing the single bulge, or in a base pair adjacent to the pair closing the bulge loop. Supplementary Table S5A provides the stabilities determined by optical melting and Supplementary Table S6A shows how well the stabilities are estimated with our nearest neighbor parameters.

We conclude from these tests that the nearest neighbor parameters are accurate enough to be used for RNA secondary structure prediction[42,43]. With the exception of one duplex with m⁶A bulged, the estimates for the duplex stabilities are within the uncertainties propagated for the experiment and the nearest neighbor parameters ($\Delta\Delta G°_{37}$ column of Supplementary Table S6A). The unmethylated consensus duplex (GGACUA-GUCC₂) is estimated by nearest neighbor parameters to be more stable (by $-0.48 \pm 0.73$ kcal/mol) than it is by experiment. The methylated consensus duplex is estimated by nearest neighbors to be less stable than it is (by $0.85 \pm 0.97$ kcal/mol). These deviations are 2.9 and 5.5% of the experimentally determined values. The estimated stability of the duplex with the dangling m⁶A closely matches the experimental value ($\Delta\Delta G°_{37}$ of $0.01 \pm 0.84$ kcal/mol).

The bulge loops closed by m⁶A-U pairs are closely modeled by the nearest neighbor parameters with $\Delta\Delta G°_{37}$ of $0.05 \pm 0.84$, $0.07 \pm 0.82$, and $0.39 \pm 0.87$ kcal/mol. For the two duplexes with bulged m⁶A, the duplexes are systematically less stable than the analogous bulged A duplex. The nearest neighbor rules do not consider the sequence identity of single bulged nucleotides, and therefore modeling this is outside the current functional form[33]. In one case, the stability of the bulged A is more accurately modeled by the sequence-independent stability and in one case, the stability of the bulged m⁶A is more accurately modeled. This points to a limitation in the nearest neighbor parameters, which has previously been documented for single nucleotide bulge loops[47,48]. In the worst case, the nearest neighbor rules are incorrect by 1.12 kcal/mol, which is only 9.4% of the total folding free energy change of the duplex. For the duplex with an m⁶A-U base pair adjacent to a G-C pair that closes a bulge loop, the agreement is also excellent, with $\Delta\Delta G°_{37}$ of $0.01 \pm 0.94$ kcal/mol.

Our optical melting experiments to derive the nearest neighbor parameters were performed in 1 M Na⁺ to be consistent with the experiments used to derive the Turner nearest neighbor parameters[33]. To test whether m⁶A is stabilized by Mg²⁺, we also performed experiments in 150 mM K⁺ and 5 mM Mg²⁺, chosen to mimic physiological salt conditions for monovalent and divalent cations[49]. Supplementary Table S5B shows the optical melting data for the Mg²⁺-containing buffer for the consensus duplex and for the eight bulge loop sequences.

We find similar folding stabilities between 1 M Na⁺ and the Mg²⁺-containing buffer (Supplementary Table S6B). The largest difference in stability was observed for the consensus methylation site duplex at $-1.79 \pm 0.84$ kcal/mol more stability in 1 M Na⁺, a difference of 12.3% between the two buffers. The bulge loop-containing duplexes were found to have no systematic bias in stability between methylated and unmethylated sequences. Prior tests of optical melting experiments in similar Mg²⁺-containing buffers generally demonstrated similar folding stability as 1 M Na⁺[50–55], with the Loop E motif in 0.1 M Na⁺ and 50 mM Mg²⁺ as a notable exception[56]. A study of hairpin stem loops with m⁶A showed a magnesium dependence on relative folding stability with and without methylation, using a relatively low monovalent salt of 25 mM NaCl and 15 mM sodium phosphate[45]. In the higher monovalent K⁺ we used, which is similar to intracellular conditions, this methylation-dependent change in stability is not observed.

**RNAstructure software modifications**. To predict RNA secondary structures for sequences with A, C, G, U, and m⁶A, we modified the command line programs in the RNAstructure software package to accept extended alphabets of nucleotides[40]. By default, the software interprets sequences as standard RNA, but a command line switch can specify an alternative alphabet. For example, the nearest neighbor parameters for a DNA alphabet composed of A, C, G, and T has long been available. Now, because of this work, the nearest neighbor parameters for an RNA m⁶A alphabet are available.

The key to an extended alphabet is the specification of the nucleotides and base pairs (Supplementary Fig. S2). A common architecture across the RNAstructure programs means that the command line programs are capable of using the extended alphabets, which can include any number of characters. This includes the prediction of minimum free energy structures, base pair probabilities, maximum expected accuracy structures, and folding stability for structures. Each nucleotide must be encoded by a single-character, and we chose "6" or "M" as the characters to encode m⁶A in sequences and in the m⁶A nearest neighbor parameter tables. The Methods section details our estimates for the m⁶A nearest neighbor parameters. The parameter tables are provided as Supplementary Materials.

Nearest neighbor parameter tables are read from disk as programs start. Each parameter table requires additional rows and columns to provide the nearest neighbor parameters values for those nucleotides, although the dimensionality of the tables stays the same. For example, a base pair stack table is four-dimensional because the sequence of four positions is required to estimate the stacking stability of two base pairs. When m⁶A is included with RNA, the size of each dimension is increased to five from four. The largest table is the $2 \times 2$ internal loop lookup table[33], which is eight dimensional because it includes the sequence of the two closing base pairs.

**Modeling the accessibility changes in MALAT1 as a result of methylation**. It has been established that N⁶-methylation can alter protein binding accessibility. To test our m⁶A nearest neighbor parameters and software, we made a quantitative prediction for the accessibility of protein binding to a structure in the lncRNA MALAT1 that is the binding target of heterogeneous nuclear ribonucleoprotein C (HNRNPC). This has been characterized by Tao Pan and co-workers in an in vitro system with a single stem-loop structure[57]. Filter binding experiments demonstrated that the methylated RNA is more accessible to protein

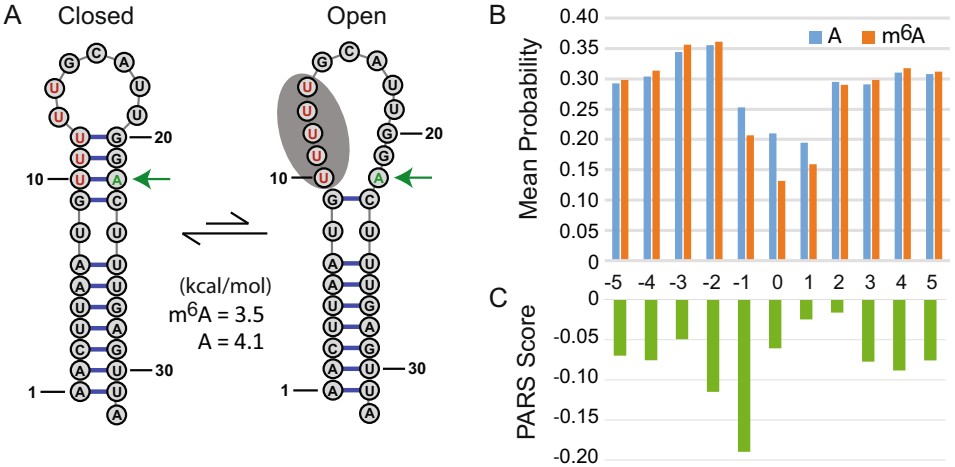

**Fig. 4 Tests of the m⁶A nearest neighbor parameters and RNAstructure software. A** The structure of MALAT1 RNA. The predicted secondary structure for the HNRNPC binding site is the closed conformation both with and without N⁶-methylation at A22 (green arrow). This is also supported by the NMR NOESY walk (Supplementary Fig. S4) and the similar chemical shifts for imino proton resonances with and without methylation (Supplementary Fig. S6). We model the binding of HNRNPC protein as the conformation that exposes the recognition sequence (marked in red nucleotides). When A22 is methylated to m⁶A, we estimate the cost of opening the binding site is reduced by 0.6 kcal/mol as compared to the unmethylated sequence. **B** The average probability that A or m⁶A are buried in a helix at the position of high-confidence m⁶A sites in the human transcriptome. The mean probability that an A or m⁶A is base paired and stacked between two adjacent pairs for 18,026 sites of N⁶-methylation, as estimated by RNAstructure. Position 0 is the site of methylation. N⁶-methylation is estimated to further open the structure at the methylation site. **C** The average PARS scores for accessibility for the 18,026 sites of N⁶-methylation in the human transcriptome. Lower PARS scores indicate higher counts of nuclease S1 cleavage relative to nuclease V1 cleavage and therefore a higher likelihood of being unpaired. The RNAstructure predictions and the PARS data both show considerable single-stranded character at the site of N⁶-methylation.

binding than the unmethylated RNA. Additionally, enzymatic cleavage by RNase S1, which has specificity for loop regions of RNA, demonstrated increased cleavage 5′ and 3′ to the methylated A, supporting increased accessibility.

We used RNAstructure to predict the lowest free energy structure of the 32 nucleotide RNA. We predicted the structure to be a hairpin stem-loop as previously modeled for the unmethylated sequence (Fig. 4A, closed structure)[57]. However, our structure prediction was unchanged for the methylated sequence. For HNRNPC to bind, the site composed of the five Us starting at position 10 must be unpaired[58]. We can therefore estimate the accessibility as the $\Delta G°_{37}$ for the breaking of the three base pairs adjacent to the hairpin loop (Fig. 4A, open structure)[59]. The $\Delta\Delta G°_{37}$ is the difference in the accessibility with and without the N⁶-methylation at position 22 is $0.6 \pm 0.4$ kcal/mol (where the uncertainty is propagated from the uncertainties of two nearest neighbor stacks that change from the introduction of the methylation). This is consistent with the $\Delta\Delta G°_{37}$ of $1.26 \pm 0.22$ kcal/mol determined by filter binding assay[57].

Our model therefore is that the N⁶-methylation of the MALAT1 hairpin does not open the binding site, but instead facilitates the HNRNPC protein-mediated breaking of base pairs at the binding site. To test this, we probed the 32 nucleotide RNA (extended with a 3′ structural cassette) by chemical mapping with CMCT (1-cyclohexyl-3-(2-morpholinoethyl) carbodiimide metho-p-toluenesulfonate), DMS (dimethyl sulfate), and kethoxal (Supplementary Fig. S3). The chemical mapping data and the prior enzymatic mapping[57] data are consistent with the proposed structure, and only minor differences are observed with and without the methylation. We also collected NMR spectra with and without methylation and assigned the imino resonances, following previous NMR studies on the same sequence[60]. The NOESY walk (Supplementary Fig. S4) confirms the hairpin conformation, and Supplementary Fig. S5 shows that a number of NMR peaks are observed at high RNA concentration because of duplex formation. Supplementary Fig. S6 also shows that the imino resonance

chemical shifts are largely unchanged in a higher Na⁺ buffer (95 mM NaCl and 38 mM sodium phosphate) as a result of N⁶-methylation, although chemical shift changes close to the methylation site are observed because of the proximity of the methyl group (Supplementary Table S7). Additionally, we tested whether Mg²⁺ alters the conformation of the N⁶-methylated sequence differently than the unmethylated sequence. The Mg²⁺ titration in Supplementary Fig. S7 shows that the imino resonances are similarly affected for the two sequences. The U10 imino resonance shifts by ~0.2 ppm in both sequences, and this was resolved by NOESY spectra (Supplementary Fig. S8). Taken together, the modeling and experimental data show that the MALAT1 RNA conformation is not directly switched by N⁶-methylation. Instead, the methylation poises the RNA to be more accessible to HNRNPC binding.

**Transcriptome-wide predictions with m⁶A**. To further test our m⁶A nearest neighbor parameters and software, we predicted structures for 18,026 mRNAs that were identified as having N⁶A methylation by whole transcriptome sequencing[61] and for which PARS structure mapping data are available[62]. We used the nearest neighbor parameters and RNAstructure package to estimate the probability that the methylation site is buried in a helix, i.e., in a base pair stacked between two other base pairs, for both the unmethylated and methylated sequence (Fig. 4B). We used 800 nucleotide fragments of local sequence to estimate the pairing probability because we previously found that pairing probability estimates for 800 nucleotide fragments closely match those for global secondary structure prediction[28]. This is a reasonable balance between accuracy and total calculation time.

We find that the unmethylated A at the methylation site is less likely to be buried in a helix than adjacent nucleotides (Fig. 4B). This is intuitive because adjacent nucleotides at the consensus site are often G or C, and A is more predominant in loops in RNAs with known structure. There is a substantial shift in the

probability of m6A being buried in a helix relative to A (21% for A and 13% for m6A). This suggests there could be widespread structural switching being affected by N6-methylation. We can also compare our results to PARS data for the same sequences (Fig. 4C)[36,62]. A PARS score quantifies the enzymatic cleavage estimate of local pairing and the experiment is performed transcriptome wide. A lower PARS score indicates greater nuclease S1 cleavage relative to nuclease V1 and thus a greater extent of unpairing because nuclease S1 has specificity for loops and nuclease V1 has specificity for helices[63]. The PARS scores at the methylation site also demonstrate a propensity to be unpaired at the methylation site, but the minimum average PARS score is at the nucleotide 5′ to the m6A site. A possible explanation for the discrepancy is that PARS attributes S1 cleavages to the base 5′ to the cleavage site, assuming that the base 5′ to the cleavage is unpaired. Cleavage can also occur when the base 3′ to the cleavage site is unpaired and therefore the PARS scores 5′ to the methylation site might be overestimating the propensity of being unpaired, in that some of the propensity of being unpaired should be attributed to the methylation site. For example, the prior S1 mapping of 5S rRNA structure is consistent with cleavages both 5′ and 3′ to unpaired nucleotides (Supplementary Fig. S9)[64]. Prior analysis of PARS scores for methylation sites also concluded that the data indicate the m6A is positioned in structures at the transition between base paired regions and loop regions, consistent with our structure prediction estimates[36].

## Discussion

Here we provide the complete nearest neighbor model for a folding alphabet including modified nucleotides. Because m6A is considered the most abundant modification in mRNA and is known to affect folding stability, we chose m6A as the modification to study. The full nearest neighbor model for secondary structure prediction requires both helical stack parameters and also loop parameters. We know from a sensitivity analysis of secondary structure prediction that, for loops, accurate parameters are most important for dangling ends and terminal mismatches[42,43], accordingly we focused our experimental effort on these motifs. We also observed marginal differences in stability for hairpin and internal loops containing m6A as compared to the same sequences without the N6-methylation. Subsequent studies could be focused on understanding and modeling folding stability differences for loops with m6A.

The other component of this study was advancing RNAstructure to work with sequences with nucleotides beyond A, C, G, and U. We provide command line tools that are ready to make quantitative predictions of structure and folding stability for sequences with m6A. The software is available for free download under the GNU GPL 2 at https://rna.urmc.rochester.edu/RNAstructure.html. Given the software, we plan expand our work in the future to include alphabets with inosine and pseudouridine. Both have helical nearest neighbor parameters available for stacks on Watson–Crick pairs[37,38], and both could be extended to full nearest neighbor parameters sets with additional optical melting experiments. The updated software also makes it possible to take advantage of engineered[65,66] or natural[67] modifications in designs of sequences for nanostructures or catalysts and a wide range of applications in biomedicine and biotechnology[68].

The two loops studied here with N6-methylations both had marginally less folding stability than the analogous unmethylated loops (Supplementary Table S4). Solution structures are available for each of the A-containing loops, and these structures provide clues as to why the stabilities would be only marginally changed

by methylation. The hairpin loop, GGCGUAAUAGCC, has the first A in the loop (A6; the site of our m6-methylation) stacked at the apex of the loop on the adjacent A (A7)[69]. Because A6 is not hydrogen bonding in the structure, a methylation at N6 can be accommodated in the preferred syn orientation by the structure without change[70]. For the internal loop with tandem G-A pairs, the pairs are trans-sugar-Hoogsteen pairs, i.e., the N6 position of the A is hydrogen bonded with the G at the N3 position[71]. For each methylated A, one hydrogen of N6 is available to form this hydrogen bond, placing the methyl in the preferred syn orientation[70]. However, the second hydrogen of A N6 is close to O4′ of the G (ranging from 2.34 to 3.36 Å in the 15 deposited NMR models). A crystal structure with this motif shows that N6 methylation disrupts pairing, with the A N6 hydrogen bonding to the O2′ of the opposite G[72]. Interestingly, this loss of pairing cost only 0.33 kcal/mol of folding stability, suggesting that conformational rearrangements do not necessarily result in a large loss of folding stability.

Recent studies demonstrated the ability of computational methods to estimate folding free energy changes[73–78]. In this work, we performed optical melting experiments to determine the folding stabilities of small model systems with m6A and fit nearest neighbor parameters to these data. Future work, however, could rely on computation or a mixture of computation and experimentation. Hopfinger et al., for example, estimated helical stacking nearest neighbor parameters for the eight stacks with m6A-U pairs adjacent to Watson–Crick pairs[73]. Overall the agreement of their estimates against our experimental values is excellent, with a root mean squared deviation of 0.30 kcal/mol. The largest single deviation is for a U-m6A pair followed by a G-C pair, where their estimate overstabilized the stack by 0.6 kcal/mol (Supplementary Fig. S10). Loop folding stabilities continue to be more of a challenge to estimate using computational methods because the conformational flexibility requires extensive sampling[76].

With this work, we demonstrate the position of m6A in a structure determines whether folding stability is increased, decreased, or unchanged relative to the same structure with A. It was previously known that N6-methylation of an A-U pair in the middle of a helix would decrease the helix folding stability[34,36]. Our stacking parameters now quantify this sequence-dependent change (Fig. 2A). It was also previously known that m6A stacking on the end of a helix would stabilize the helix more than an analogous A. In this work, we also discovered that an m6A-U base pair at the terminal position of a helix provides roughly the same folding stability as an analogous A-U base pair. This is because terminal A-U pairs destabilize helices with a penalty of +0.45 kcal/mol[44] that is not needed for terminal m6A-U base pairs (Fig. 2B). Recently, it was also discovered that terminal G-U base pairs in helices do not need an end penalty[79]. These results, taken together, show why N6-methylation is a potent switch of secondary structure.

Our transcriptome-wide calculations also suggest that structure switches from N6-methylation might be widespread (Fig. 4B). It will be interesting to perform similar calculations with other widespread covalent modifications, such as inosine. There is potential to identify structural mechanisms by which covalent modifications exert changes in protein binding, transcript stability, or gene expression.

## Methods

**Synthesis of oligonucleotides with m6A.** Oligoribonucleotides were synthesized on a BioAutomation MerMade12 DNA/RNA synthesizer using β-cyanoethyl phosphoramidite chemistry and commercially available RNA phosphoramidites (ChemGenes, GenePharma) and protected N6-methyladenosine phosphoramidite,

which was synthetized according to a standard protocol. Synthesis of N[6]-methyladenosine was conducted via Dimroth rearrangement followed by protection of the 5′-hydroxyl with dimethoxytrityl and 2′-hydroxyl with tert-butyldimethylsilyl. Next, 5′- and 2′-protected N[6]-methyladenosine was treated with 2-cyanoethyl N,N,N′,N′-tetraisopropylphosphorodiamidite[80,81]. Oligoribonucleotides were deprotected with aqueous ammonia/ethanol (3/1 v/v) for 16 h at 55 °C. Silyl protecting groups were cleaved by treatment triethylamine trihydrofluoride. Deprotected oligonucleotides were purified by silica gel thin layer chromatography in 1-propanol/aqueous ammonia/water (55/35/10 v/v/v)[44,81].

**Optical melting data**. The thermodynamic measurements were performed for nine various concentrations of RNA duplex in the range of 0.1 mM to 1 μM in buffer containing 1 M sodium chloride, 20 mM sodium cacodylate, and 0.5 mM Na$_2$EDTA, pH 7. A subset of additional optical melting experiments (Supplementary Table S5B) was performed in a buffer containing 150 mM KCl, 20 mM cacodylic acid, and 5 mM MgCl$_2$, pH corrected to 7 using KOH. Oligonucleotide single strand concentrations were calculated from the absorbance above 80 °C and single strand extinction coefficients were approximated by a nearest neighbor model[82]. Absorbance vs. temperature melting curves were measured at 260 nm with a heating rate of 1 °C/min from 0 to 90 °C on JASCO V-650 spectrophotometer with a thermoprogrammer. The melting curves were analyzed and the thermodynamic parameters calculated from a two-state model with the program MeltWin 3.5[52]. For most model RNAs, the ΔH° derived from $T_M^{-1}$ vs. ln($C_T$/a) plots, where a is 4 for non-self-complementary and a is 1 for self-complementary duplexes, is within 15% of that derived from averaging the fits to individual melting curves, as expected if the two-state model is reasonable. Supplementary Fig. S11 shows the UV absorbance as a function of temperature and the $T_M^{-1}$ vs. ln($C_T$/a) plots for 5′CGGUGCm[6]AUCG$_2$ in a buffer with 1 M NaCl, 5′GGCAGm[6]ACUC/3′ CCGCUGAG in a buffer with 1 M NaCl, and 5′GGCAGm[6]ACUC/3′CCGCUGAG in a buffer with 150 mM KCl and 5 mM MgCl$_2$.

**Linear regression**. Linear least-squares fitting to determine RNA stacking stabilities was performed with a custom Python program using the statsmodels ordinary least-squares class[83]. For each duplex, to determine the stabilities to be fit, the fixed terms were subtracted, including the stability of base pair stacks with Watson–Crick and G-U pairs only, the duplex initiation term, the terminal A-U penalty term (when needed), and the symmetry term (when needed). The free energy changes from the $T_M^{-1}$ vs. ln($C_T$/a) fits of the optical melting data were used. The fit was excellent, with coefficient of determination, $R^2$, of 0.984. Uncertainty estimates (Fig. 2A and Supplementary Table S2) are the standard errors of the regression. Supplementary Table S8 shows the stability to be fit and the estimate of the fit. Supplementary Table S9 shows the number of occurrences of each stacking parameter in the set of fit helices. The Python code for fitting nearest neighbor parameters is provided as Supplementary Software 1.

**Loop motif stability calculations**. Loop motif stabilities (Supplementary Table S4) are calculated by subtracting the helical component of stability. The free energy changes from the $T_M^{-1}$ vs. ln($C_T$/a) fits of the optical melting data were used.

For the dangling ends and terminal mismatches, twice the stability increment of the motif is determined by subtracting a reference helix stability from the stability of the duplex with the motif:

$$2 \times \Delta G^\circ_{37\,motif} = \Delta G^\circ_{37\,duplex\,with\,two\,motifs} - \Delta G^\circ_{37\,reference\,duplex\,without\,the\,motifs} \quad (1)$$

The factor of two is present because the self-complementary duplexes have two instances of the motif.

For the hairpin loop, the stability of the loop motif is determined by subtracting the stability of the helical stacks (estimated with nearest neighbor parameters) from the total stability:

$$\Delta G^\circ_{37\,hairpin\,loop} = \Delta G^\circ_{37\,stem-loop} - \Delta G^\circ_{37\,helical\,stacks} \quad (2)$$

The total helical stack stability is reported as the Reference $\Delta G^\circ_{37}$ in Supplementary Table S4.

For the internal loop, the stability is the total stability of the duplex minus the helical stacks and helix initiation (estimated with nearest neighbor parameters) and minus the stability cost of symmetry (because the duplex is self-complementary):

$$\Delta G^\circ_{37\,internal\,loop} = \Delta G^\circ_{37\,duplex\,with\,internal\,loop} - \Delta G^\circ_{37\,helical\,stacks} - \Delta G^\circ_{37\,initiation} - \Delta G^\circ_{37\,symmetry} \quad (3)$$

The Reference $\Delta G^\circ_{37}$ reported in Supplementary Table S4 is the sum of the helical stacks and symmetry free energy increments.

**Error propagation**. To estimate uncertainties in free energies (σ), we propagate uncertainty estimates for experiments and nearest neighbor parameters using the standard method for uncorrelated parameters:

$$\sigma^2 = \sum_i \left( \sigma_i \frac{\partial \Delta G^\circ}{\partial \Delta G^\circ_i} \right)^2 \quad (4)$$

where ΔG°$_i$ is the $i$th term (or experimental value) and $σ_i$ is the uncertainty in the $i$th term[43]. For the sum of terms used here, this simplifies to:

$$\sigma^2 = \sum_i \left( n_i \sigma_i \right)^2 \quad (5)$$

where $n_i$ is the number of occurrences of the $i$th parameter or the $i$th experiment. For uncertainty estimates for optical melting experiments, we use 4% of the magnitude of the ΔG°$_{37}$. This was chosen as a conservative estimate of the precision of optical melting by Xia et al.[44]. It is twice the mean difference in free energies determined using the two fit methods for optical melting data (Average of Curve Fits and Analysis of $T_M$ Dependence) for a database of optical melting experiments.

**Nearest neighbor parameter determination**. Nearest neighbor parameters were developed to estimate the folding stability (ΔG°$_{37}$) of sequences with A, C, G, U, and m[6]A. Nearest neighbor parameters are inherited from the 2004 Turner Rules[33], for which a summary of their derivation can be found in Zuber et al.[42] and examples for their use are available on the Nearest Neighbor Database website. Helical stacking tables are from Xia et al.[44] for Watson–Crick stacks and from Chen et al.[79] for stacks that contain G-U pairs, supplemented with the stacks determined for m[6]A-U pairs in this work. Following Chen et al., terminal G-U base pairs in a helix are not penalized.

Dangling end m[6]As are stabilized as compared to the analogous A dangling end by the mean additional stability found here (−0.4 kcal/mol). Dangling ends on m[6]A-U pairs are assumed to be the same stability as dangling ends on A-U base pairs. When the stability is measured by an experiment, the measured value is used (Supplementary Table S4).

Terminal mismatches involving m[6]A are estimated to be more stable than the analogous A terminal mismatch by the mean value found for the terminal mismatches in this study (−0.3 kcal/mol). M[6]A-m[6]A terminal mismatches receive only −0.3 kcal/mol additional stability. A terminal mismatch on an m[6]A-U pair is also stabilized by −0.3 kcal/mol compared to the analogous mismatch on an A-U pair. These effects are additive; an m[6]A-containing terminal mismatch on an m[6]A-U pair receives an additional −0.6 kcal/mol stability than the analogous terminal mismatch with all A parameters. When the stability is measured by an experiment, the measured value is used (Supplementary Table S4).

Hairpin, internal, and bulge loop initiation costs are length-dependent[33]. The same length-dependent costs are used here, which assumes that m[6]A does not alter the initiation costs.

In total, 1 × 1, 2 × 1, and 2 × 2 internal loop stabilities are stored in lookup tables. The stabilities for loops with unpaired m[6]A are taken from the analogous loops with A. And m[6]A-U-closed loops are taken from analogous A-U-closed loops with one change. A-U-closed loops have a 0.7 kcal/mol stability penalty per closure[33]; for m[6]A-U-closed loops, this cost has been removed compared to the analogous A-U-closed loop. Larger internal loops use a terminal stacking table to assign a stability increment for the sequence of the closing pair and first mismatch. Separate tables are used for loops of size 1 × n, 2 × 3, and (>2) × (>2)[33]. These terminal stack tables use the analogous A parameter for stacks with m[6]A. The one exception is that the +0.7 kcal/mol internal loop A-U pair closure penalty is removed for m[6]A-U closures.

Hairpin loop tables for triloop, tetraloops, and hexaloops are unchanged. These tables include stabilities for specific hairpin sequences known by experiment to not be well predicted using nearest neighbor rules[33]. Other hairpin loops are estimated with the sum of a terminal mismatch and a length-dependent initiation. The terminal mismatches for m[6]A use the analogous A parameter.

Multibranch loop initiation parameters are from an experimental fit using a simple linear model. Coaxial stacking is included in multibranch and exterior loops[33]. Coaxial stacking between two adjacent helices is assumed to be as stable as a helical stack. For coaxial stacks with an intervening mismatch, there are two stacks. The coaxial stacking increment for the stack where the backbone is not continuous was previously found to be independent of sequence, and the sequence-independent value is used here for stacks involving one or more m[6]As. The other stack is identical to the terminal mismatch stack table.

**Extended alphabet implementation in RNAstructure**. RNAstructure is a software package written in C++, with a C++ class library that is also wrapped using SWIG to be available to JAVA or Python programs[40]. It is open source and provided for free under the GNU GPL license version 2 at https://rna.urmc.rochester.edu. A number of the command line programs have been updated to be capable of using extended alphabets, including Fold (secondary structure prediction by free energy minimization), efn2 (estimation of folding free energy changes for secondary structures), and partition (partition function calculations for estimating pair, motif, or structure probabilities). A number of programs that rely on the partition function calculations are therefore also able to consider extended alphabets, including design (design of a sequence to fold to a specific secondary structure), EnsembleEnergy (calculation of the ensemble folding free energy change), MaxExpect (prediction of maximum expected accuracy structures), ProbKnot (prediction of structures that can include pseudoknots), ProbScan (estimation of motif probabilities), and stochastic (stochastic sampling from the Boltzmann ensemble).

The underlying secondary structure prediction algorithms remain unchanged, although substantial changes were made to expand the software from the four nucleotide nucleic acid alphabet to a fully user-customizable folding alphabet. To do this, the command line tools read the thermodynamic parameters at startup. The switch "--alphabet" is used to specify the set of parameters to be used. The default is "rna", the current (2004) Turner rules for estimating RNA folding free energy changes[33,44]. Included with the latest RNAstructure release (version 6.3) is also "m6A", the parameters discussed here, and "dna", a set of nearest neighbor rules for DNA secondary structure prediction. The files are a plain text format that was updated (in version 6.0) for extended alphabets. The specification file (Supplementary Fig. S2) is read first, and it defines the alphabet and base pairs. Dynamic memory allocation is used to provide the memory needed to store the tables. The parameters themselves are then read from the files.

The 2004 Turner rules gave a terminal base pair penalty for any base pair (A-U or G-U) at the end of a helix that contained a U[33,44]. In this work, we found that terminal m6A-U pairs did not require this terminal base pair penalty. Additionally, the revised G-U stack parameters[79], used with the m6A parameters we derived, do not require a terminal base pair penalty. Therefore, we changed the implementation of the energy function to account for this change. The default "rna" rules are unchanged and still penalize terminal G-U base pairs.

**Chemical mapping of RNA and data analysis**. DMS (to modify adenosine and cytidine), CMCT (to modify uridine and guanosine) and kethoxal (to modify guanosine) were used to chemically map the secondary structure of the 32 nucleotide RNA (with a 3′-structural cassette). The RNA (5′AACUUAAU-GUUUUUGCAUUGGACUUUGAGUUACCUUCCGGGCUUCGGUCCG GAAC) was synthesized using the phosphoramidite method on a MerMade synthesizer, deprotected, and purified on a 12% denaturing gel. The RNA contained a structure cassette at the 3′ end (underlined), which was designed using RNAstructure to fold independently and allow readout of whole structure of studied RNA. The RNA contained C16-2′-OMe instead of a standard C nucleotide at position 16, introduced to prevent nonenzymatic spontaneous cleavage between C16 and A17[84]. For each reaction, 10 pmol of RNA was folded in buffer containing 300 mM NaCl, 10 mM Tris-HCl, 5 mM MgCl2 pH 8.0. Briefly, the appropriate amount of RNA was diluted in H2O and heated 3 min in 80 °C followed by slow cooling. Then, at 50 °C a concentrated buffer was added to the final buffer solution and the sample was continuously slowly cooled. After 10 min incubation at 4 °C, chemical mapping was conducted using two concentrations of each reagent. To a 9 µl sample, 1 µl of 300 mM or 160 mM DMS in ethanol was added to give a final concentration of 30 or 15 mM DMS. For modification with CMCT, 9 µl of CMCT solution was added to the 9 µl of RNA sample. CMCT was diluted in a folding buffer to give a final concentration of 250, and 100 mM in the reaction mixture. Kethoxal was diluted in ethanol/water (1:3 v/v) to give a final concentration of 160 and 80 mM. After modification with kethoxal, 3 µl of 35 mM potassium borate solution was added to stabilize the products of modification. Chemical modification reactions were incubated for 1.5 h at 4 °C. Reactions were stopped by precipitation with ethanol. The chemical modification reactions were repeated for a total of two replicates of each agent. The RNA in control reactions were treated the same, except no chemical reagents were added.

Modification sites were identified by primer extension. The DNA primer for reverse transcription (RT) was synthesized with 6-fluorescein (FAM) on the 5′ end (5′FAMGTTCCGGACCGAAGCCCG). The DNA primer was complementary to 3′ end of RNA (the cassette part). For each RT reaction, 10 pmol of primer was used. Primer extension was performed at 55 °C with SuperScript III reverse transcriptase using Invitrogen's protocol. Reactions were stopped by addition of loading buffer containing urea and 10 mM EDTA, then chilling on ice. Prior to separation and readout of cDNA products, the samples were heated for 5 min at 95 °C and then separated on a 12% polyacrylamide denaturing gel (Supplementary Fig. S12).

The gel image from the Phosphorimager was analyzed using the SAFA program to quantify nucleotide reactivities[85]. cDNA products were identified by comparing to sequencing lanes and to control lanes, and the raw results from SAFA were normalized. To quantify chemical modification at each nucleotide, we first corrected for the background by subtracting the volume integral of the band in the control lane from the volume integral of experimental lane. For each of two experiments for each modification agent and each sequence, we characterized the modification extent by quartiles. When a nucleotide was in the highest quartile of RT stops in both experiments, we report the mapping as strong (Supplementary Fig. S3). When a nucleotide was in the second highest quartile in both experiments or the highest quartile in one and the second highest quartile in the other, we report the cleavage as moderate.

**NMR**. Methylated and unmodified RNA samples were exchanged into desired buffer for NMR spectroscopy by two 10-fold cycles of dilution followed by centrifugal concentration using Amicon Ultra-4 filter units with 3000 MW cutoff. Final volume was brought to 0.52 ml, including 4% D2O.

NMR spectra were collected with a Varian Inova 600 MHz spectrometer with a standard (RT) triple-resonance triple-axis gradient probe. One dimensional and two dimensional (2D) NMR spectra in 96% H2O/4% D2O were collected using a WATERGATE pulse[86,87] with flipback for water suppression at indicated temperatures. NOESY and TOCSY spectra were acquired at 10 °C. NOESY spectra

with excitation optimized for imino protons were acquired with mixing times of 50, 150, and 250 ms, and with excitation optimized for aromatic protons and greater indirect resolution with mixing times of 75 and 400 ms. TOCSY spectra (30 ms mixing time) were used to identify pyrimidine H5-H6 cross-peaks and dynamic sugar residues. Spectra from 2D NMR were processed and assigned with NMRpipe[88] and NMRFAM-SPARKY[89] as described[90].

**Transcriptome-wide calculations**. We downloaded the set of m6A positions reported in the human transcriptome by Schwartz et al.[61], which was available as their Supplementary Table S2. Using a Python program, for each entry for the human genome of "high-confidence category" and with a RefSeq entry, we fetched the sequence from RefSeq[91] using the Bio.Entrez module from Biopython[92]. To identify the exact position of the m6A in the transcript, we used the provided hg19 coordinates to identify the A in one of the expected sequence motifs (GGACA, GGACT, GGACC, GAACT, AGACT, AGACA, or TGACT) using the twobitreader Python package and the hg19 sequence downloaded from the UCSC genome browser[93]. Once the motif was identified in the genome, the sequence was found in the RefSeq sequence, and an 800 nucleotide FASTA sequence was generated with the m6A position at the 401st position. For sequences in which the m6A was too close to the 5′ end or 3′ end to be in the 401st position, up to 800 nucleotides were extracted with the m6A position at either the 5′ end or 3′ end. Sequences were generated with both A and 6 at the m6A position. In total, 18,155 high-confidence m6A sites were found.

Next, the partition function was calculated for each 800 nucleotide sequence using the partition program from RNAstructure[40]. To determine the probability that the m6A position was buried in a helix, a custom C++ program was written using inheritance of the RNA class[40]. The probability of the $i$th nucleotide being buried in a helix is the sum for all $j$ of the probability the $i$-$j$ base pair is sandwiched between the base pairs $(i-1)$-$(j+1)$ and $(i+1)$-$(j-1)$. Each of these can be determined using the partition function, Q, as a normalization factor and partial partition function for interior and exterior fragments.

$$P_i = \sum_{j=1}^{N} V'(i-1, j+1) \times K_{stack}(i-1, j+1, i, j) \times K_{stack}(i, j, i+1, j-1) \times V(i+1, j-1) \tag{6}$$

where $P_i$ is the probability that nucleotide $i$ is buried in a helix, $N$ is the length of the sequence, $V'(i,j)$ is the partition function for the exterior fragment of nucleotides 1 to $i$ to $j$ and to $N$ given that $i$ is paired to $j$, $K_{stack}(i,j,i+1,j-1)$ is the equilibrium constant for the base pair stack of base pairs $i$-$j$ and $(i+1)$-$(j-1)$, and $V(i,j)$ is the partition function for the interior fragment from nucleotides $i$ to $j$ given that $i$ is paired to $j$. Supplementary Fig. S13 diagrams $V'$ and $V$. These arrays of partition functions for sequence fragments are also explained in a description of the partition function calculation[94].

**PARS calculations**. To calculate PARS scores for human transcripts, we downloaded the dataset deposited by Wan et al.[62] to the NCBI GEO (Gene Expression Omnibus)[95]. We used the mapped reads available for S1-treated (accessions GSM1226157, GSM1226159, and GSM1226161) and V1-treated (accessions GSM1226158, GSM1226160, and GSM1226162) samples. We calculated the PARS score using[62]:

$$PARS_i = \log_2 \left[ \frac{V1_i \times \left( \frac{S1_{total}}{V1_{total}} \right) + 5}{S1_i + 5} \right] \tag{7}$$

where $PARS_i$ is the PARS score for the $i$th nucleotide, $V1_i$ is the number of reads in the V1-treated samples attributed to the $i$th nucleotide, $S1_i$ is the number of reads in the S1-treated samples attributed to the $i$th nucleotide, $S1_{total}$ is the total number of $S1$-treated sample reads, and $V1_{total}$ is the total number of $V1$-treated sample reads. The ratio of $S1_{total}$ and $V1_{total}$ is a normalization factor. The addition of 5 in the numerator and denominator is a pseudocount to reduce the magnitude of scores for positions with few reads[62]. In total, entries were found for 18,026 transcripts of the 18,155 high-confidence m6A-containing transcripts found.

**Reporting summary**. Further information on research design is available in the Nature Research Reporting Summary linked to this article.

## Data availability
The results of the optical melting experiments needed to reproduce the results in this study are provided in the Supplementary Materials (Supplementary Tables S1 and S3). The raw data are available from the corresponding authors upon request. The nearest neighbor parameters are available as Supplementary Materials as Supplementary Software 2 and as part of the RNAstructure software package.

## Code availability
RNAstructure is provided under the GPL V2 license, and it is therefore free and open source. It can be downloaded at http://rna.urmc.rochester.edu/RNAstructure.html. The nearest neighbor fitting code is available as Supplementary Materials.

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

## Acknowledgements

This work was supported by National Institutes of Health grant R01GM076485 to D.H.M. and by National Science Center grants: UMO-2020/01/0/NZ6/00137 to E.K. and UMO-2019/33/B/ST4/01422 to R.K. Calculations were performed at the University of Rochester Center for Integrated Research Computing.

## Author contributions

E.K. designed experiments, synthesized strands, performed optical melting experiments, performed chemical mapping experiments, and revised the manuscript, X.Z. contributed code to RNAstructure, R.M.W. contributed code to RNAstructure, S.D.K. performed and analyzed the NMR experiments, M.S. performed optical melting experiments, R.K. synthesized phosphoramidites and strands and revised the manuscript, D.H.M. designed experiments, contributed code to RNAstructure, fit the nearest neighbor parameters, and drafted the manuscript.

## Competing interests

The authors declare no competing interests.
