## [Peer Review File · Nature Communications]

Reviewers' Comments:

Reviewer #1:

Remarks to the Author:

Although increasing attention has been paid to covalently modified nucleotides in RNAs, computational methods for predicting RNA secondary structures including such modified nucleotides are still lacking. m6A is the most popular modification and thus worth incorporating into a structural analysis. The authors present a new version of RNAstructure software that can accept m6A nucleotides as input. The algorithm makes use of nearest neighbor parameters that consider m6A, which are determined by their own optical melting experiments. For a biological application, they discussed effects of N6-methylation by predicting subsets of mRNAs and lncRNAs that are known to contain m6A.

The major contribution of this paper is to provide a method for predicting secondary structures of RNA sequences including m6A nucleotides with a full set of the corresponding nearest neighbor parameters, as well as giving insights into the thermodynamic stability of structures that contain m6A. I agree that it is important for the community to consider the secondary structure analysis of RNA sequences including m6A nucleotides. However, it seems to me that the algorithmic framework is the same as the early version of RNAstructure. Unfortunately, there seems to be no novelty in terms of the algorithm itself. In my impression, the updates are accepting extended alphabets of nucleotides and incorporating new thermodynamic parameters determined experimentally in this work to the dynamic programming algorithm. This is a simple extension of their earlier work that can predict RNA secondary structures of standard sequences, though this new version can deal with RNA sequences including m6A. An input RNA sequence is assumed to contain m6A and to know their positions in their setting, but in reality how they can detect with a computational approach is also an important task for true usability. I would like the authors to address this point. The authors discussed the thermodynamic stability of structures that contain m6A, but I think their conclusions of conformational switching are still at a starting point and need additional examination to give an impact on RNA biology.

Minor

-There are many typos and grammatical errors throughout the manuscript. For example, the first paragraph in Introduction on page 3 includes the phrase "now can now." In the third paragraph on the same page, "the" should be followed by the phrase "chicken-and-egg problem." I will not raise here these kinds of errors one by one.

-Table S1

Please define ΔG , ΔH and ΔS .

Reviewer #2:

Remarks to the Author:

Summary: The authors perform optical melting experiments (UV absorbance measurements at 260 nm for 9 distinct concentrations of 32 sort RNA sequences) to determine the thermodynamic parameters for m6A-U base pair stacking, and terminal mismatches.

General comments: It was a real pleasure to read such a carefully written account of work, carried out to such perfection. The authors are to be commended on such superlative work. The computational extension to RNAstructure is/was straightforward, but necessary to make available the MFE structure prediction, partition function computation, etc. for sequences that include N6-methylated adenines.

A devil's advocate might argue that the significance of the extension is not great, since the improvement in prediction accuracy has not been assessed, apart from corroboration with PARS data on a semi-quantitative manner. Concrete suggestions to address this issue are given below.

Specific comments:

1) Four of the 15 values of free energy change ($\mu \pm \sigma$) for stacked pairs involving 6-methyl A found in Table S2 have absolute values $|\mu|$ of about 0.3 kcal/mol or less, with the property that $\sigma > |\mu|$. Thus arguably, a quarter of the base stacking free energies involving m6A-U pairs are not statistically different than zero.

This raises the question of significance of this otherwise extremely well carried out work. The authors make a good argument concerning one conformational switch (MALAT1 RNA) triggered by N6-methylation of A. As well, the authors show by computation that 800 nt fragments of lncRNAs (whose methylation profile is known) that m6A is less likely to be buried in a helix, which is loosely corroborated by their analysis of PARS data: "The PARS scores at the methylation site also demonstrate a propensity to be unpaired at the methylation site" (page 7)

A way to possibly strengthen the impact of the authors' results would be to predict the structure of those transfer RNAs from the tRNAdb (<http://trnadb.bioinf.uni-leipzig.de/DataOutput/> repository for an extension of the Sprinzl database), that contain m6A. By comparing predictions using RNAstructure both with and without the extended alphabet, the authors could compute sensitivity and PPV values. This work is straightforward and easily carried out, so I suggest the authors add these results to the paper. Admittedly, there are usually only 5 or 6 instances of modified nucleotides (often pseudo-uridine) in tRNAs, that there may be no difference in structure prediction with/without accounting for N6-methylation of adenines. In that case, perhaps there could be a stronger difference in the base pairing probabilities computed for base pairs known to belong to the native structure.

2) On page 4, in the 3rd paragraph of "Results", the word "melting" is misspelled by "meting".

3) According to the authors, the extension of RNAstructure uses the usual treatment of special hairpin loops, such as the tetraloops CAACGG and CCAAGG. It would be interesting if the authors could carry out a small number of additional experiments to determine whether N6-methylation of the adenines in (for instance) these two examples (or others the authors can stipulate) induces a significant change in the free energies. For the first instance (CAACGG), the A in position 2 is a mismatch, hence handled by the authors' treatment in the paper; the effect of methylation on the A in position 3 as well may be insignificant. However, there may be a larger difference in CCAAGG or other examples, in which case the authors could strongly suggest that enthalpy (single-stranded base stacking in hairpins) is the reason for the bonuses.

4) It would be useful for the Mathews Lab to add the new parameters to the NNDB for easy public access.

Reviewer #3:

Remarks to the Author:

The manuscript 'Secondary Structure Prediction for RNA Sequences Including N6-methyladenosine' describes for the first time a method for RNA structure prediction using modified bases. The work is of high significance to the field and defines a new era in RNA structure prediction towards the ever growing number of modified bases. Given that RNAstructure has a long tradition and is one of the most widely used tools in this field, this publication is expected to have major impact.

The methodology is sound and meets the expected standards in the field. All material is publicly available (the source code is published under GPL), additional analysis and calculations are described in great detail and are fully reproducible.

Minor comments:

*) p3 par1: ...'now can now be localized transcriptome-wide...' remove first 'now'

*) The material in 'Reviewer Instructions.tar' should be made available to readers as supplementary information.

We appreciate the feedback from our reviewers. Here all reviewer comments are copied verbatim and appear italicized. We respond to each suggestion in roman type.

Reviewer #1 (Expertise: RNA structural prediction):

Although increasing attention has been paid to covalently modified nucleotides in RNAs, computational methods for predicting RNA secondary structures including such modified nucleotides are still lacking. m⁶A is the most popular modification and thus worth incorporating into a structural analysis. The authors present a new version of RNAstructure software that can accept m⁶A nucleotides as input. The algorithm makes use of nearest neighbor parameters that consider m⁶A, which are determined by their own optical melting experiments. For a biological application, they discussed effects of N⁶-methylation by predicting subsets of mRNAs and lncRNAs that are known to contain m⁶A.

The major contribution of this paper is to provide a method for predicting secondary structures of RNA sequences including m⁶A nucleotides with a full set of the corresponding nearest neighbor parameters, as well as giving insights into the thermodynamic stability of structures that contain m⁶A. I agree that it is important for the community to consider the secondary structure analysis of RNA sequences including m⁶A nucleotides. However, it seems to me that the algorithmic framework is the same as the early version of RNAstructure. Unfortunately, there seems to be no novelty in terms of the algorithm itself. In my impression, the updates are accepting extended alphabets of nucleotides and incorporating new thermodynamic parameters determined experimentally in this work to the dynamic programming algorithm. This is a simple extension of their earlier work that can predict RNA secondary structures of standard sequences, though this new version can deal with RNA sequences including m⁶A.

We appreciate that the reviewer agrees that the community will benefit from the ability to predict secondary structures that account for changes because of N⁶-methylation of A.

The reviewer is correct that the underlying algorithms in RNAstructure are unchanged. For the software aspects we report, the importance is in the software engineering; this includes developing the input/output formats that can handle any user-customizable alphabet and the internal representations of sequence and thermodynamic parameters. The changes to the RNAstructure package were substantial.

To better emphasize the scope and importance of this, we revised the section “Extended Alphabet Implementation in RNAstructure” in the Methods to clarify this. We also changed the text in the caption of Figure 1.

Additionally, we want to point to the fact that the determination of nearest neighbor rules for the folding of sequences including m⁶A was a substantial effort that was only possible with our collaboration. 43 new optical melting experiments were performed with one or more m⁶A nucleotides.

An input RNA sequence is assumed to contain m⁶A and to know their positions in their setting, but in reality how they can detect with a computational approach is also an important task for true usability. I

would like the authors to address this point. The authors discussed the thermodynamic stability of structures that contain m6A, but I think their conclusions of conformational switching are still at a starting point and need additional examination to give an impact on RNA biology.

We agree with the reviewer that identifying sites of N⁶-methylation is also an important problem. In this manuscript, however, we focus on structure prediction when the sites of m⁶A are known. In our opinion, identifying methylation sites is a distinct problem. This problem is addressed experimentally by available sequencing methods¹⁻⁶, it is addressed computationally by recent work focused on the problem of predicting sites of N⁶-methylation⁷⁻¹⁰, and it is addressed with a new database of known methylation sites¹¹.

Minor

-There are many typos and grammatical errors throughout the manuscript. For example, the first paragraph in Introduction on page 3 includes the phrase "now can now." In the third paragraph on the same page, "the" should be followed by the phrase "chicken-and-egg problem." I will not raise here these kinds of errors one by one.

We fixed these errors and additional errors we found with careful proofreading.

-Table S1

Please define delta G, delta H and delta S.

We defined the terms ΔH° , ΔS° , and ΔG°_{37} in the legend for Table S1.

Reviewer #2 (Expertise: RNA structural prediction):

Summary: The authors perform optical melting experiments (UV absorbance measurements at 260 nm for 9 distinct concentrations of 32 sort RNA sequences) to determine the thermodynamic parameters for m6A-U base pair stacking, and terminal mismatches.

General comments: It was a real pleasure to read such a carefully written account of work, carried out to such perfection. The authors are to be commended on such superlative work. The computational extension to RNAstructure is/was straightforward, but necessary to make available the MFE structure prediction, partition function computation, etc. for sequences that include N6-methylated adenines.

A devil's advocate might argue that the significance of the extension is not great, since the improvement in prediction accuracy has not been assessed, apart from corroboration with PARS data on a semi-quantitative manner. Concrete suggestions to address this issue are given below.

We thank the reviewer for the assessment of the work. We appreciate that the reviewer found it a pleasure to read the work. We also agree that the assessment of accuracy is incomplete. In our

opinion, it has been difficult to develop hypotheses about the effect of methylation on structure and that is why the literature has few detailed structure analyses in spite of widespread evidence that methylation is important. It is our hope that the advance we report will facilitate developing and testing of hypotheses.

Specific comments:

1) Four of the 15 values of free energy change ($\mu \pm \sigma$) for stacked pairs involving 6-methyl A found in Table S2 have absolute values $|\mu|$ of about 0.3 kcal/mol or less, with the property that $\sigma > |\mu|$. Thus arguably, a quarter of the base stacking free energies involving m6A-U pairs are not statistically different than zero.

The reviewer made an interesting observation about the stacking values and their uncertainties. The stacking nearest neighbor model is rooted in the biophysical model that base pairing and stacking both contribute to the stabilization of helices. Because we fit to this physical model, stability values close to zero are also meaningful.

This raises the question of significance of this otherwise extremely well carried out work. The authors make a good argument concerning one conformational switch (MALAT1 RNA) triggered by N6-methylation of A. As well, the authors show by computation that 800 nt fragments of lncRNAs (whose methylation profile is known) that m6A is less likely to be buried in a helix, which is loosely corroborated by their analysis of PARS data: "The PARS scores at the methylation site also demonstrate a propensity to be unpaired at the methylation site" (page 7)

A way to possibly strengthen the impact of the authors' results would be to predict the structure of those transfer RNAs from the tRNADB (<http://trnadb.bioinf.uni-leipzig.de/DataOutput/> repository for an extension of the Sprinzl database), that contain m6A. By comparing predictions using RNAstructure both with and without the extended alphabet, the authors could compute sensitivity and PPV values. This work is straightforward and easily carried out, so I suggest the authors add these results to the paper. Admittedly, there are usually only 5 or 6 instances of modified nucleotides (often pseudo-uridine) in tRNAs, that there may be no difference in structure prediction with/without accounting for N6-methylation of adenines. In that case, perhaps there could be a stronger difference in the base pairing probabilities computed for base pairs known to belong to the native structure.

The reviewer has an excellent idea. We followed up on this suggestion by predicting the secondary structures for all 24 sequences in the Sprinzl tRNA database that are reported to contain m⁶A (represented with the "=" symbol)¹². We generated the accepted structure from the available alignment, but additionally added the helices in the variable loop regions of serine, leucine, and tyrosine tRNAs. These helices are absent in the Sprinzl database, but the location of these helices was guided by the genomic tRNA database available from Todd Lowe¹³.

We mapped the modified nucleotides other than m⁶A back to typical RNA alphabet. For those bases that cannot pair in an A-form helix (because of disruption of the Watson-Crick face or because the base is not planar), we forced the nucleotide to be unpaired in structure prediction. This is consistent with

prior benchmarks of tRNA structure prediction¹⁴. We additionally experimented with allowing all modified nucleotides to pair and the change did not alter our conclusions.

The minimum free energy structures were unchanged with and without the knowledge of the N⁶-methylation sites. On average, for these 24 sequences, the sensitivity (the percent of known pairs correctly predicted) was 76.1%. The positive predictive value (PPV; the percent of predicted pairs that are correct) was 74.6%.

We also calculated the normalized ensemble defect (NED) for folding to the accepted structure with and without methylation¹⁵. The NED is the average fraction of nucleotides in the Boltzmann ensemble that are estimated to not fold in the accepted structure. Therefore, an NED of 0 would be a perfect prediction for the accepted structure and an NED of 1 would mean that no pairs in the accepted structure were predicted to occur in the ensemble. Here we observe a change in favor of using our new m⁶A parameters. The mean NED when using A in place of m⁶A was 0.307. The mean NED when using m⁶A was 0.299. The P value, however, from a one-tailed, paired t-test was 0.063. Therefore, we conclude that the knowledge of m⁶A thermodynamics trends towards improved prediction accuracy, but that the improvement is not statistically significant. It would be helpful to have additional known structures that included m⁶A because that could provide more power for the comparison.

We chose to not include these predictions as part of the manuscript because the findings were not significant.

2) On page 4, in the 3rd paragraph of "Results", the word "melting" is misspelled by "meting".

We fixed two instances of this misspelling.

3) According to the authors, the extension of RNAstructure uses the usual treatment of specials hairpin loops, such as the tetraloops CAACGG and CCAAGG. It would be interesting if the authors could carry out a small number of additional experiments to determine whether N6-methylation of the adenines in (for instance) these two examples (or others the authors can stipulate) induces a significant change in the free energies. For the first instance (CAACGG), the A in position 2 is a mismatch, hence handled by the authors' treatment in the paper; the effect of methylation on the A in position 3 as well may be insignificant. However, there may be a larger difference in CCAAGG or other examples, in which case the authors could strongly suggest that enthalpy (single-stranded base stacking in hairpins) is the reason for the bonuses.

The reviewer asks interesting questions. We did not extend the tetraloop tables (composed of tetraloops with folding stabilities greater than the nearest neighbor rules would otherwise predict) with loops with m⁶A in place of A. This was the conservative choice, and the impact of additional experiments might be a revision of the tetraloop tables.

At this time, we would prefer to not do these experiments. One important reason is that we used the work of Zuber et al. to guide our experiments to determine the values for parameters that are likely to have the most impact for secondary structure prediction accuracy^{16,17}. In that work, we characterized the extent to which fluctuations in parameter values alter structure predictions (quantified as root mean

squared deviations of base pairing probabilities). That work demonstrated that tetraloop (and triloop and hexaloop) parameters have little impact on precision of secondary structure prediction. We therefore think it is better to focus on single mismatches and terminal mismatches, which have substantially larger impacts on structure prediction accuracy. An additional reason to not do the experiments at this time is that there is a strong consensus sequence for sites of N⁶-methylation that does not overlap with our special tetraloop sequences like CAACGG and CCAAGG. These loop sequences could be important in engineering applications where m⁶A is inserted in a designed sequence, but they are unlikely to be biologically important.

We hypothesize that the N⁶-methylation of either A in CCAAGG would be unlikely to substantially change the folding stability compared to A. The structure of this tetraloop is available in a ribosome crystal structure (residues 414 to 419 in pdb accession #4U4R¹⁸) and is nicely diagrammed in Figure 3E of D'Ascenzo et al.¹⁹. Like the methylation site of the hairpin loop studied in our manuscript, CGU6AUAG, the two As in the tetraloop can accommodate an N⁶-methyl in the syn position. The N⁶ of A416 is not hydrogen bonded. The N⁶ of A417 is hydrogen bonded to the non-bridging oxygen of the phosphate for A416, but the hydrogen engaged is in the anti position. We did not find a crystal structure with the tetraloop CAACGG, and therefore we are not able to hypothesize how N⁶-methylation would change its folding stability.

4) It would be useful for the Mathews Lab to add the new parameters to the NNDB for easy public access.

The NNDB has extensive explanations and examples for the use of the nearest neighbor parameters. It will take us some time to add the necessary web pages for the m⁶A-alphabet parameters. We agree, however, that it would be helpful for the community for the parameter tables to be more easily available than just with the full RNAstructure software bundle. To address this, we added a zip file of the parameters and a README file to explain their use as a supplementary materials.

Reviewer #3 (Expertise: non-coding RNA biology, structural prediction):

The manuscript 'Secondary Structure Prediction for RNA Sequences Including N6 - methyladenosine' describes for the first time a method for RNA structure prediction using modified bases. The work is of high significance to the field and defines a new era in RNA structure prediction towards the ever growing number of modified bases. Given that RNAstructure has a long tradition and is one of the most widely used tools in this fields, this publication is expected to have major impact.

The methodology is sound and meets the expected standards in the field. All material is publicly available (the source code is published under GPL), additional analysis and calculations are described in great detail and are fully reproducible.

We thank the reviewer for the appreciation of this work.

Minor comments:

**) p3 par1: ... 'now can now be localized transcriptome-wide...' remove first 'now'*

We fixed this mistake and other typos we found with careful proofreading.

**) The material in 'Reviewer Instructions.tar' should be made available to readers as supplementary information.*

The information in 'Reviewer Instructions.tar' (about how to predict secondary structures with RNAstructure) was drawn from our online tutorial at:

<https://rna.urmc.rochester.edu/tutorials/workshop/index.html>

This is part of the online help we provide for RNAstructure, and therefore we prefer to not include it as Supplementary Information.

Additional change to the manuscript:

In addition to changes to the manuscript in response to reviewer feedback, we also substantially changed our analysis of the MALAT1 hairpin stem loop. In our prior manuscript, we hypothesized that methylation led to a conformational change, on average, in the folding ensemble. We had tested that hypothesis with chemical mapping, which did not provide strong evidence either for or against our model.

To better test the MALAT1 model we continued to study the hairpin after submitting our manuscript. We collected NMR spectra on the stem-loop both with and without N⁶-methylation at position 22 (Figure 4A). The NMR data demonstrate that the stem-loop does not change conformation in response to methylation (Figures S4, S5, and S6). This means that the methylation facilitates the opening of the protein binding site (red Us in Figure 4A) without opening the pairs in the absence of the protein, and a more appropriate analysis is the change in folding free energy for opening the nucleotides. Changes in the Results and Figure 4A reflect our improved understanding of the MALAT1 hairpin. Supplemental Table S7 and Figures S4, S5, and S6 are added to provide the NMR data. Scott Kennedy collected and analyzed the NMR data and he was added as a coauthor.

References:

- 1 Schwartz, S. *et al.* High-resolution mapping reveals a conserved, widespread, dynamic mRNA methylation program in yeast meiosis. *Cell* **155**, 1409-1421 (2013).
- 2 Linder, B. *et al.* Single-nucleotide-resolution mapping of m6A and m6Am throughout the transcriptome. *Nat Methods* **12**, 767-772 (2015).
- 3 Ke, S. *et al.* A majority of m6A residues are in the last exons, allowing the potential for 3' UTR regulation. *Genes Dev* **29**, 2037-2053 (2015).
- 4 Dominissini, D. *et al.* Topology of the human and mouse m6A RNA methylomes revealed by m6A-seq. *Nature* **485**, 201-206 (2012).
- 5 Meyer, K. D. *et al.* Comprehensive analysis of mRNA methylation reveals enrichment in 3' UTRs and near stop codons. *Cell* **149**, 1635-1646 (2012).
- 6 Chen, K. *et al.* High-resolution N(6) -methyladenosine (m(6) A) map using photo-crosslinking-assisted m(6) A sequencing. *Angew Chem Int Ed Engl* **54**, 1587-1590 (2015).

- 7 Dao, F. Y. *et al.* Computational identification of N6-methyladenosine sites in multiple tissues of mammals. *Comput Struct Biotechnol J* **18**, 1084-1091 (2020).
- 8 Zhou, Y., Zeng, P., Li, Y. H., Zhang, Z. & Cui, Q. SRAMP: prediction of mammalian N6-methyladenosine (m6A) sites based on sequence-derived features. *Nucleic Acids Res* **44**, e91 (2016).
- 9 Zhang, Y. & Hamada, M. DeepM6ASeq: prediction and characterization of m6A-containing sequences using deep learning. *BMC Bioinformatics* **19**, 524 (2018).
- 10 Xiang, S., Liu, K., Yan, Z., Zhang, Y. & Sun, Z. RNAMethPre: A Web Server for the Prediction and Query of mRNA m6A Sites. *PLoS One* **11**, e0162707 (2016).
- 11 Tang, Y. *et al.* m6A-Atlas: a comprehensive knowledgebase for unraveling the N6-methyladenosine (m6A) epitranscriptome. *Nucleic Acids Res* **49**, D134-D143 (2021).
- 12 Juhling, F. *et al.* tRNADB 2009: compilation of tRNA sequences and tRNA genes. *Nucleic Acids Res* **37**, D159-162 (2009).
- 13 Chan, P. P. & Lowe, T. M. GtRNADB 2.0: an expanded database of transfer RNA genes identified in complete and draft genomes. *Nucleic Acids Res* **44**, D184-189 (2016).
- 14 Mathews, D. H., Sabina, J., Zuker, M. & Turner, D. H. Expanded sequence dependence of thermodynamic parameters provides improved prediction of RNA secondary structure. *J Mol Biol* **288**, 911-940 (1999).
- 15 Zadeh, J. N., Wolfe, B. R. & Pierce, N. A. Nucleic acid sequence design via efficient ensemble defect optimization. *J Comput Chem* **32**, 439-452 (2011).
- 16 Zuber, J., Cabral, B. J., McFadyen, I., Mauger, D. M. & Mathews, D. H. Analysis of RNA Nearest Neighbor Parameters Reveals Interdependencies and Quantifies the Uncertainty in RNA Secondary Structure Prediction. *RNA* **24**, 1568-1582 (2018).
- 17 Zuber, J., Sun, H., Zhang, X., McFadyen, I. & Mathews, D. H. A sensitivity analysis of RNA folding nearest neighbor parameters identifies a subset of free energy parameters with the greatest impact on RNA secondary structure prediction. *Nucleic Acids Res* **45**, 6168-6176 (2017).
- 18 Garreau de Loubresse, N. *et al.* Structural basis for the inhibition of the eukaryotic ribosome. *Nature* **513**, 517-522 (2014).
- 19 D'Ascenzo, L., Leonarski, F., Vicens, Q. & Auffinger, P. Revisiting GNRA and U-NCG folds: U-turns versus Z-turns in RNA hairpin loops. *RNA* **23**, 259-269 (2017).

Reviewers' Comments:

Reviewer #1:

Remarks to the Author:

I agree that determining nearest neighbor rules for the reported case is substantial, which would be useful for future energy-based tools with the same purpose. The additional experiment on the MALAT1 hairpin stem loop with NMR is convincing and can give us full insights into the accessibility of the RNA recognition model. I believe this revision could enhance the authors' presentation.

Reviewer #2:

Remarks to the Author:

The authors have met all the concerns of the reviewers, with the exception of of the concern about significance and impact of the results. The article is very well written, and the authors have done additional work, including NMR studies (with an additional author). The results are important and the community should become aware of this extension of RNAstructure. In summary, the article is solid, well-executed, and is comparable to the many contributions of the Turner Lab in their papers that appeared in the journal Biochemistry, etc.

With all this said, however, in my mind, there is still a concern whether the results are sufficiently significant for publication in Nature Communications. There is no algorithmic novelty, although the reviewer is aware of the additional labor in modifying the code to handle the new input. Unfortunately, such time-consuming efforts are often unrewarded yet necessary when releasing new versions of software. Moreover, it has been clear to this reviewer for 10-15 years, and certainly clear as well to the authors and to RNA computational researchers that it is necessary ultimately to modify existent data structures (in Vienna Package, RNAstructure, NUPACK, etc.) to handle additional types of base pairs (ultimately for non-canonical base pairing). However, in my mind, such additional work, though necessary, does not increase the merit of a paper unless it also increases the significance and impact of the paper.

Reviewer #3:

Remarks to the Author:

The authors carefully addressed all concerns raised by the referees, provided new analysis and subjected the text to careful proofreading. Overall, the quality of the manuscript has significantly improved upon revision.

The additional NMR analysis of MALAT1 provides important insights on how RNA modifications may affect structured regions, even if the destabilizing effects are minimal.

Reviewer #4:

Remarks to the Author:

Kierzek et al developed a method to quantitatively predict the secondary structure of m6A containing RNAs. The experiments and computation are of high quality as is typical of the Mathews lab. While understanding the impact of m6A on conformational thermodynamics is an important goal for studies of RNA folding and function, the current manuscript is too narrow in two important respects which significantly diminishes its physiological relevance.

First, all of the UV melts and NMR experiments were performed under non-physiological conditions in the absence of Mg²⁺ ions. Not only are Mg²⁺ ions important for RNA folding, stability, and function, but there are also studies in the literature (e.g. Liu et al Nat Comm 2018) showing that the impact of m6A on RNA structure and stability can be strongly dependent on Mg²⁺. Second, the manuscript is primarily focused on a narrow set of motifs – namely helical elements, in which m6A has already been shown to have a destabilizing effect. However, the manuscript does not address other very common RNA motifs, in which m6A has been shown to have the opposite effect,

stabilizing RNA structure in a context and Mg²⁺ dependent. For example, m⁶A has been shown to destabilize RNA when placed in a junctional bp right next to a 3' bulge but to stabilize RNA when placed next to a 5' bulge. Although the authors assessed the effect of m⁶A on terminal bps, they did not consider helix-junction-helix motifs, which are widespread in RNAs. Indeed, there is evidence to suggest most physiological m⁶A are in such motifs (Liu et al., 2018). It is therefore unlikely that the current approach will successfully predict the effects of m⁶A on even common RNA secondary structures. The conclusion regarding how m⁶A affects MALAT1 RNA is also questionable; except for chemical probing, all experiments were also performed in the absence of Mg²⁺. Additionally, the NMR data reported in this paper are not new and are almost identical to those reported previously (Zhou et al., 2016) and the latter paper is not cited. Finally, I agree with reviewers 1 and 2 that the study lacks novelty with regards to the algorithm itself.

I believe that to have physiological relevance, at least for a subset of motifs, the data needs to be repeated in the presence of Mg²⁺ to verify key conclusions and trends, and the motifs should be expanded to include at minimum the bulge motif in which m⁶A is likely to be found in vivo. Some of the latter data could even be obtained from the literature. Alternatively, the authors should provide a compelling argument for why these points are not a concern when it comes to modeling RNA secondary structure under physiological conditions.

Minor comments:

The authors concluded that m⁶A does not switch the structure of MALAT1 but rather increase the accessibility for opening of the RNA. The usage of "opening" may not be accurate here as base opening (Guéron et al., 1987) and melting are two different concepts. The author may refer to local melting of the duplex region in MALAT1 in this context.

It would be helpful to show representative UV melting curves for different motifs to demonstrate the quality of optical melting data and whether m⁶A leads to deviations from two-state transitions.

Another key paper (Huang et al., 2017) regarding the impact of m⁶A on tandem A-G mismatches was not cited in this manuscript.

References

- Guéron, M., Kochoyan, M., and Leroy, J. L.: A single mode of DNA base-pair opening drives imino proton exchange, *Nature*, 328, 89-92, 10.1038/328089a0, 1987.
- Huang, L., Ashraf, S., Wang, J., and Lilley, D. M.: Control of box C/D snoRNP assembly by N⁶-methylation of adenine, *EMBO Rep*, 18, 1631-1645, 10.15252/embr.201743967, 2017.
- Liu, B., Merriman, D. K., Choi, S. H., Schumacher, M. A., Plangger, R., Kreutz, C., Horner, S. M., Meyer, K. D., and Al-Hashimi, H. M.: A potentially abundant junctional RNA motif stabilized by m⁶A and Mg²⁺, *Nat Commun*, 9, 2761, 10.1038/s41467-018-05243-z, 2018.
- Zhou, K. I., Parisien, M., Dai, Q., Liu, N., Diatchenko, L., Sachleben, J. R., and Pan, T.: N⁶-Methyladenosine Modification in a Long Noncoding RNA Hairpin Predisposes Its Conformation to Protein Binding, *J Mol Biol*, 428, 822-833, 10.1016/j.jmb.2015.08.021, 2016.

Here all reviewer comments are copied verbatim and appear italicized. We respond to each suggestion in roman type. We appreciate this additional reviewer feedback, and we made use of the feedback to improve our manuscript.

Reviewer #1 (Expertise: RNA structural prediction):

I agree that determining nearest neighbor rules for the reported case is substantial, which would be useful for future energy-based tools with the same purpose. The additional experiment on the MALAT1 hairpin stem loop with NMR is convincing and can give us full insights into the accessibility of the RNA recognition model. I believe this revision could enhance the authors' presentation.

We thank the reviewer for these positive comments.

Reviewer #2 (Expertise: RNA structural prediction):

The authors have met all the concerns of the reviewers, with the exception of of the concern about significance and impact of the results. The article is very well written, and the authors have done additional work, including NMR studies (with an additional author). The results are important and the community should become aware of this extension of RNAstructure. In summary, the article is solid, well-executed, and is comparable to the many contributions of the Turner Lab in their papers that appeared in the journal Biochemistry, etc.

With all this said, however, in my mind, there is still a concern whether the results are sufficiently significant for publication in Nature Communications. There is no algorithmic novelty, although the reviewer is aware of the additional labor in modifying the code to handle the new input. Unfortunately, such time-consuming efforts are often unrewarded yet necessary when releasing new versions of software. Moreover, it has been clear to this reviewer for 10-15 years, and certainly clear as well to the authors and to RNA computational researchers that it is necessary ultimately to modify existent data structures (in Vienna Package, RNAstructure, NUPACK, etc.) to handle additional types of base pairs (ultimately for non-canonical base pairing). However, in my mind, such additional work, though necessary, does not increase the merit of a paper unless it also increases the significance and impact of the paper.

We thank the reviewer for commenting that the work is important, that it is useful for the community, and that it is comparable to the many contributions from the Turner lab.

We believe the work is more significant than the reviewer gives it credit. The substantial amount of software development work required to support the extended alphabets is complemented by substantial thermodynamic experiments to develop the m⁶A thermodynamic parameters. This opens many new possible avenues of exploration of secondary structure for natural modifications and for synthetic nucleotides, including modeling structures and for designing nanostructures. We added text to the end of the Discussion to explain the importance of synthetic nucleotides and engineered structures.

Reviewer #3 (Expertise: non-coding RNA biology, structural prediction):

The authors carefully addressed all concerns raised by the referees, provided new analysis and subjected the text to careful proofreading. Overall, the quality of the manuscript has significantly improved upon revision.

The additional NMR analysis of MALAT1 provides important insights on how RNA modifications may affect structured regions, even if the destabilizing effects are minimal.

We appreciate the favorable comments from the reviewer.

Reviewer #4 (new reviewer; expertise NMR characterization of RNA structure):

Kierzek et al developed a method to quantitatively predict the secondary structure of m6A containing RNAs. The experiments and computation are of high quality as is typical of the Mathews lab. While understanding the impact of m6A on conformational thermodynamics is an important goal for studies of RNA folding and function, the current manuscript is too narrow in two important respects which significantly diminishes its physiological relevance.

First, all of the UV melts and NMR experiments were performed under non-physiological conditions in the absence of Mg²⁺ ions. Not only are Mg²⁺ ions important for RNA folding, stability, and function, but there are also studies in the literature (e.g. Liu et al Nat Comm 2018) showing that the impact of m6A on RNA structure and stability can be strongly dependent on Mg²⁺.

We chose to perform the melts in this study in 1 M Na⁺. This is key for developing nearest neighbor parameters for m⁶A that work with the 2004 Turner rules^{1,2}, which were derived from optical melting studies performed in 1 M Na⁺. The large majority of optical melting experiments have been performed in 1 M Na⁺ and in the absence of Mg²⁺. In part, this choice is historical because Mg²⁺ induces strand cleavage at high temperature; the RNA strands for model systems were historically much more difficult to obtain. The other reason 1 M Na⁺ has remained the mainstay for these studies is that melts obtained on model RNA systems in a more physiological salt condition, ~150 mM K⁺ and ~5-10 mM Mg²⁺, have generally similar stability to 1 M Na⁺³⁻⁸. A notable exception to this is the Loop E motif, which in 100 mM Na⁺ and 50 mM Mg²⁺ is more stable than in 1 M Na⁺⁹.

Although the nearest neighbor rules (a.k.a. Turner Rules) are derived in 1 M Na⁺, they provide accurate predictions for RNA secondary structures *in vitro* and in cells. The accuracy of predictions can be improved by using multiple homologous sequences^{10,11} or by using experimental mapping data in the calculations¹².

We clarified the importance of our choice of 1 M Na⁺ in the Results section.

Second, the manuscript is primarily focused on a narrow set of motifs – namely helical elements, in which m6A has already been shown to have a destabilizing effect. However, the manuscript does not address other very common RNA motifs, in which m6A has been shown to have the opposite effect, stabilizing RNA structure in a context and Mg²⁺ dependent. For example, m6A has been shown to destabilize RNA when placed in a junctional bp right next to a 3' bulge but to stabilize RNA when placed

next to a 5' bulge. Although the authors assessed the effect of m6A on terminal bps, they did not consider helix-junction-helix motifs, which are widespread in RNAs. Indeed, there is evidence to suggest most physiological m6A are in such motifs (Liu et al., 2018). It is therefore unlikely that the current approach will successfully predict the effects of m6A on even common RNA secondary structures. The conclusion regarding how m6A affects MALAT1 RNA is also questionable; except for chemical probing, all experiments were also performed in the absence of Mg²⁺.

As the reviewer states, the focus of experiments is on helices, which were shown in our prior work to be the single most important terms for precise secondary structure prediction. We did, however, study an internal loop (helical junction), which is reported in Table S3. We also studied a number of dangling ends and terminal mismatches (Table S3), which are needed for extrapolation loop terms. Other terms in the nearest neighbor model are inherited from the 2004 Turner Rules, and precise secondary structure prediction was shown to be less dependent on the precision of these terms^{13,14}.

Additionally, the NMR data reported in this paper are not new and are almost identical to those reported previously (Zhou et al., 2016) and the latter paper is not cited.

We appreciate that the reviewer pointed out that we did not cite Zhou et al. (2016). We added that reference, and we expanded our NMR results to point out differences in peak assignments. Our conclusions are very similar to the previous work. An important aspect of our NMR work was that we identified a number of the peaks at high sample concentration are a result of duplex formation. By diluting the NMR sample, we were able to remove the duplex peaks. This was important for demonstrating that one hairpin species occurs, and no competing single-stranded species are present.

For this revision, we added additional NMR data taken in the presence of Mg²⁺ (Supplemental Figures S7 and S8). Based on these data, we concluded that the presence or absence of the methylation did not substantially change the structure in the presence of Mg²⁺. We performed a titration as a function of Mg²⁺ because the duplex species was substantially stabilized by the addition of Mg²⁺ to our relatively low Na⁺ concentration (that was chosen to diminish the duplex). The chemical shifts as a function Mg²⁺ concentration are largely unchanged for methylated and N⁶-methylated strands. However, the imino resonance for U10 does change by ~0.2 ppm with the addition of Mg²⁺ for both the methylated and unmethylated strands.

Finally, I agree with reviewers 1 and 2 that the study lacks novelty with regards to the algorithm itself.

We agree that the secondary structure predictions algorithms are unchanged. However, the RNAstructure software package was substantially revised to be able to include more than four nucleotides. It is the combination determining the set of m⁶A nearest neighbor parameters and revising the software to use these, and future parameter sets that include modified nucleotides, that makes this work novel.

I believe that to have physiological relevance, at least for a subset of motifs, the data needs to be repeated in the presence of Mg²⁺ to verify key conclusions and trends, and the motifs should be expanded to include at minimum the bulge motif in which m6A is likely to be found in vivo. Some of the latter data could even be obtained from the literature. Alternatively, the authors should provide a compelling argument for why these points are not a concern when it comes to modeling RNA secondary structure under physiological conditions.

We expanded the set of sequences we studied as test of the nearest neighbor parameters (Tables S5A and S6A) by including six bulge loops with a bulged m⁶A, a closure by an m⁶A-U base pair, or with an m⁶A-U base pair adjacent to a pair closing a bulge loop. We also did optical melting experiments on six analogous sequences with unmethylated A. We modeled four of the sequences on the hairpin stem-loops studied by Liu et al. (2018), although we used duplexes to reduce the melting temperature and to be able to collect data with sequence dependence on the melting temperature. For bulge loops closed by m⁶A-U pairs, the estimates are within 0.33 kcal/mol of the experiments. For experiments on two duplexes with bulged m⁶A, we find that the stabilities are systematically lower than for the analogous A bulge. The current Turner rules do not account for sequence dependence in bulged nucleotides, and therefore we do not account for this difference. Interestingly, the nearest neighbor parameter predictions are closer to the stability of the m⁶A bulge instead of the analogous A bulge in one of two cases. For the bulge loop with an m⁶A-U pair adjacent to the closing pair, the agreement is excellent ($\Delta\Delta G_{37}^\circ = -0.05 \pm 0.63$ kcal/mol).

We also expanded the tests to include optical melting experiments with a buffer containing Mg²⁺ and that is roughly physiological (150 mM KCl and 5 mM MgCl₂)¹⁵⁻²¹. The stabilities of 9 duplexes in the Mg²⁺-containing buffer are reported in Table S5B. Table S6B compares the stabilities to stabilities in 1 M NaCl. The differences are small (12.3% at the largest) and there are no systematic differences in stability for m⁶A as compared to analogous duplexes with A.

Minor comments:

The authors concluded that m6A does not switch the structure of MALAT1 but rather increase the accessibility for opening of the RNA. The usage of “opening” may not be accurate here as base opening (Gueron et al., 1987) and melting are two different concepts. The author may refer to local melting of the duplex region in MALAT1 in this context.

We changed the text to refer to the free energy change of “breaking” the three base pairs in the hairpin in order to make the structure accessible to protein binding. We also added a reference that details the equilibrium free energy calculation²², which we have also used in subsequent work^{23,24}.

It would be helpful to show representative UV melting curves for different motifs to demonstrate the quality of optical melting data and whether m6A leads to deviations from two-state transitions.

We now provide representative UV melting curves as Supplemental Figure S11.

Another key paper (Huang et al., 2017) regarding the impact of m6A on tandem A-G mismatches was not cited in this manuscript.

We thank the reviewer for pointing us to this reference. We now cite this reference in the Discussion section.

References

Gueron, M., Kochoyan, M., and Leroy, J. L.: A single mode of DNA base-pair opening drives imino proton exchange, *Nature*, 328, 89-92, 10.1038/328089a0, 1987.

Huang, L., Ashraf, S., Wang, J., and Lilley, D. M.: Control of box C/D snoRNP assembly by N6-methylation of adenine, *EMBO Rep*, 18, 1631-1645, 10.15252/embr.201743967, 2017.

Liu, B., Merriman, D. K., Choi, S. H., Schumacher, M. A., Plangger, R., Kreutz, C., Horner, S. M., Meyer, K. D., and Al-Hashimi, H. M.: A potentially abundant junctional RNA motif stabilized by m(6)A and Mg(2), *Nat Commun*, 9, 2761, 10.1038/s41467-018-05243-z, 2018.

Zhou, K. I., Parisien, M., Dai, Q., Liu, N., Diatchenko, L., Sachleben, J. R., and Pan, T.: N(6)-Methyladenosine Modification in a Long Noncoding RNA Hairpin Predisposes Its Conformation to Protein Binding, *J Mol Biol*, 428, 822-833, 10.1016/j.jmb.2015.08.021, 2016.

References:

- 1 Mathews, D. H. *et al.* Incorporating chemical modification constraints into a dynamic programming algorithm for prediction of RNA secondary structure. *Proc Natl Acad Sci USA* **101**, 7287-7292 (2004).
- 2 Xia, T. *et al.* Thermodynamic parameters for an expanded nearest-neighbor model for formation of RNA duplexes with Watson-Crick pairs. *Biochemistry* **37**, 14719-14735 (1998).
- 3 Diamond, J. M., Turner, D. H. & Mathews, D. H. Thermodynamics of three-way multibranch loops in RNA. *Biochemistry* **40**, 6971-6981 (2001).
- 4 Jaeger, J. A., Zuker, M. & Turner, D. H. Melting and chemical modification of a cyclized self-splicing group I intron: similarity of structures in 1 M Na⁺, in 10 mM Mg²⁺, and in the presence of substrate. *Biochemistry* **29**, 10147-10158 (1990).
- 5 McDowell, J. A. & Turner, D. H. Investigation of the structural basis for thermodynamic stabilities of tandem GU mismatches: Solution structure of (rGAGGUCUC)₂ by two-dimensional NMR and simulated annealing. *Biochemistry* **35**, 14077-14089 (1996).
- 6 Xia, T., McDowell, J. A. & Turner, D. H. Thermodynamics of nonsymmetric tandem mismatches adjacent to G-C base pairs in RNA. *Biochemistry* **36**, 12486-12487 (1997).
- 7 Schroeder, S. J. & Turner, D. H. Factors affecting the thermodynamic stability of small asymmetric internal loops in RNA. *Biochemistry* **39**, 9257-9274 (2000).
- 8 Jiang, T., Kennedy, S. D., Moss, W. N., Kierzek, E. & Turner, D. H. Secondary structure of a conserved domain in an intron of influenza A M1 mRNA. *Biochemistry* **53**, 5236-5248 (2014).
- 9 Serra, M. J. *et al.* Effects of magnesium ions on the stabilization of RNA oligomers of defined structures. *RNA* **8**, 307-323 (2002).
- 10 Seetin, M. G. & Mathews, D. H. RNA structure prediction: an overview of methods. *Methods Mol Biol* **905**, 99-122 (2012).
- 11 Havgaard, J. H. & Gorodkin, J. RNA structural alignments, part I: Sankoff-based approaches for structural alignments. *Methods Mol Biol* **1097**, 275-290 (2014).
- 12 Sloma, M. F. & Mathews, D. H. Improving RNA secondary structure prediction with structure mapping data. *Methods Enzymol* **553**, 91-114 (2015).
- 13 Zuber, J., Cabral, B. J., McFadyen, I., Mauer, D. M. & Mathews, D. H. Analysis of RNA Nearest Neighbor Parameters Reveals Interdependencies and Quantifies the Uncertainty in RNA Secondary Structure Prediction. *RNA* **24**, 1568-1582 (2018).
- 14 Zuber, J., Sun, H., Zhang, X., McFadyen, I. & Mathews, D. H. A sensitivity analysis of RNA folding nearest neighbor parameters identifies a subset of free energy parameters with the greatest impact on RNA secondary structure prediction. *Nucleic Acids Res* **45**, 6168-6176 (2017).

- 15 Leamy, K. A., Assmann, S. M., Mathews, D. H. & Bevilacqua, P. C. Bridging the gap between in vitro and in vivo RNA folding. *Q Rev Biophys* **49**, e10 (2016).
- 16 Feig, A. L. & Uhlenbeck, O. C. in *The RNA World, 2nd edn* (eds R. F. GESTELAND, T. R. CECH, & J.F. ATKINS) 287-320 (Cold Spring Harbor Press, 1999).
- 17 Alberts, B., Bray, D., Lewis, J., Roberts, K. & Watson, J. D. *Molecular Biology of the Cell, 3rd edn.* (Garland Publishing, 1994).
- 18 London, R. E. Methods for measurement of intracellular magnesium: NMR and fluorescence. *Annu Rev Physiol* **53**, 241-258 (1991).
- 19 Romani, A. M. Magnesium homeostasis in mammalian cells. *Front Biosci* **12**, 308-331 (2007).
- 20 Lusk, J. E., Williams, R. J. & Kennedy, E. P. Magnesium and the growth of Escherichia coli. *J Biol Chem* **243**, 2618-2624 (1968).
- 21 Truong, D. M., Sidote, D. J., Russell, R. & Lambowitz, A. M. Enhanced group II intron retrohoming in magnesium-deficient Escherichia coli via selection of mutations in the ribozyme core. *Proc Natl Acad Sci U S A* **110**, E3800-3809 (2013).
- 22 Mathews, D. H., Burkard, M. E., Freier, S. M., Wyatt, J. R. & Turner, D. H. Predicting oligonucleotide affinity to nucleic acid targets. *RNA* **5**, 1458-1469 (1999).
- 23 Lu, Z. J. & Mathews, D. H. OligoWalk: An online siRNA design tool utilizing hybridization thermodynamics. *Nucleic Acids Research* **36**, W104-W108 (2008).
- 24 Mathews, D. H. Using OligoWalk to Identify Efficient siRNA Sequences. *Methods in Molecular Biology* **629**, 109-121 (2010).

Reviewers' Comments:

Reviewer #4:

Remarks to the Author:

The authors have done a good job addressing my comments and concerns and the manuscript is now suitable for publication.

Here all reviewer comments are copied verbatim and appear italicized. We respond to each suggestion in roman type.

Reviewer #4:

The authors have done a good job addressing my comments and concerns and the manuscript is now suitable for publication.

We appreciate that the reviewer agrees the manuscript is ready for publication. We thank the reviewer for the detailed feedback on the prior submission, which helped us improve the manuscript.